# Rigid Amorphous Fraction as an Indicator for Polymer-Polymer Interactions in Highly Filled Plastics

**DOI:** 10.3390/polym13193349

**Published:** 2021-09-30

**Authors:** Johannes Benz, Christian Bonten

**Affiliations:** Institut für Kunststofftechnik, University of Stuttgart, Pfaffenwaldring 32, 70569 Stuttgart, Germany; christian.bonten@ikt.uni-stuttgart.de

**Keywords:** particle-particle interaction, inner structure, thermal analysis, highly filled plastic, rigid amorphous fraction, crystallization

## Abstract

Above a percolation threshold a flow restriction has to be overcome by higher pressure in plastic processing. Besides amount and geometry of fillers, the interactions of polymer and filler are important. By differing the amorphous phase of polymers into a rigid amorphous and a mobile amorphous fraction, predictions about interactions are possible. The objective is the generation of a flow restriction and the combined investigation of polymer–particle interaction. SiO_2_ was used up to 50 vol.% in different spherical sizes in PLA and PP. A capillary-rheometer was used as a tool to create a yield point and by that investigations into the state of the flow restriction were possible. All produced compounds showed, in plate-plate rheometry, an increase in viscosity for lower shear rates and a significant change in the storage modulus. In DSC, hardly any specific rigid amorphous fraction was detectable, which suggests that there is a minor interaction between macromolecules and filler. This leads to the conclusion that the change in flow behavior is mainly caused by a direct interaction between the particles, even though they are theoretically too far away from each other. First images in the state of the yield point show a displacement of the particles against each other.

## 1. Introduction

Every material melts at least once during processing, which is why the properties of the plastic melt play a decisive role. Fillers in particular, which are added to the polymer to adjust the properties of the end product or to create them in the first place, influence processing right through to the mold.

The rheological behavior of the plastic melt can be fundamentally influenced in compounding. In particular, the amount of filler is the first to be mentioned here, which is combined with the polymer. In general, it can be said that the higher the filler content, the higher the viscosity of the plastic. This is due to the fact that, with solid fillers, they cannot be sheared and thus hinder the flowability [1].

Figure 1 shows the relative viscosity of filled systems and that the viscosity of the polymer increases with increasing filler content. The figure also shows that the geometry of the filler changes the viscosity. The geometry can be expressed in two terms. The size of the filler particles and the aspect ratio, which takes into account the shape and thus also the surface of the additive particle, play an important role. The higher the surface, the more possibilities there are for interaction with the polymer.

The amount of the additive in the plastic must be considered first, since a necessary condition for a changed flow behavior is that the percolation threshold is exceeded. An increase in viscosity can already be achieved by a small addition, with the viscosity curve being comparable to that of the polymer. A further increase of the filler content may result in an increase of the viscosity, especially at low shear rates, rather than in the formation of a Newtonian plateau. This is called a yield point or at least the formation of a flow inhibition [1,2,3]. Due to the fact that particles may rub against each other, an internal structure may form in the plastic because of the high filler content [4]. Furthermore, this can be understood as an increase in the flow resistance [5,6,7], which can be seen in the increase of the storage modulus, especially for lower shear rates.

**Figure 1 polymers-13-03349-f001:**
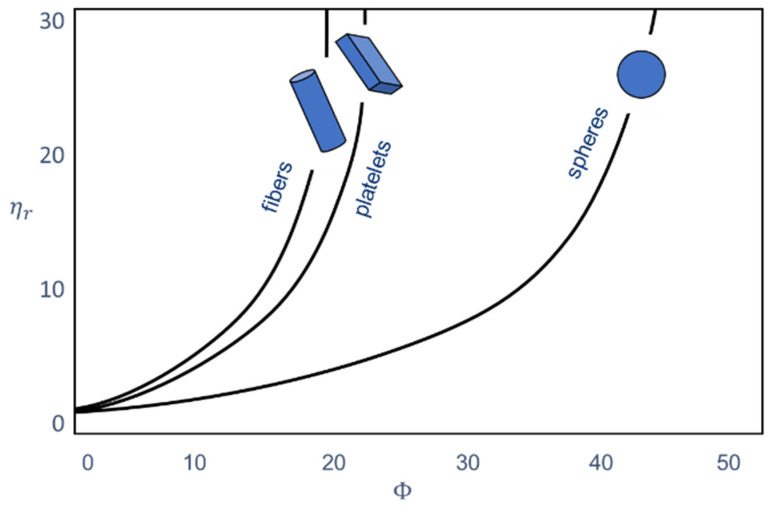
Influence of the filler content and the particle shape on the relative viscosity of plastics [5]. The relative viscosity is the ratio of the viscosity of the filled plastic to the viscosity of the unfilled plastic. The information on the filler content is based on the volume.

The cause of this internal structure can be a particle-particle interaction, a particle–polymer interaction, or their combination. A possible macromolecule-particle interaction must be determined in advance before its influence can be evaluated. One possible approach is to further refine semi-crystalline polymers into a two-phase model with subdivision into amorphous and crystalline phases (crystalline fraction, CF), as shown in Figure 2, far left. Wunderlich [8,9] and Schick [10] show the extension to a three-phase model. Here, the amorphous phase is divided into a mobile amorphous fraction (MAF) and a rigid amorphous fraction (RAF). The interface between the crystalline and the (mobile) amorphous region is formed by the macromolecules in the rigid amorphous fraction. Basically, it can be assumed that macromolecules cross both crystalline and amorphous regions due to their length. This can be understood as a covalent bond between the two phases. This influences the mobility of these macromolecules in a special way [11,12].

This can be verified using thermal analysis, since the reduced mobility results in both a decrease in the specific heat capacity and an increase in the glass transition temperature T_G_ [12]. The stronger these rigid amorphous macromolecules are bound in crystalline regions, the higher the energy input will be that is required to melt these regions. The temperature at which the rigid regions will melt has not yet been fully clarified. However, it seems that these macromolecules are only released when the crystalline regions are melted [13].

The determination of the rigid amorphous phase is done at
(1)$$\mathrm{RAF}=1-\mathrm{CF}-\frac{\Delta c_{p}}{\Delta c_{p,\mathrm{pure}}}=1-\mathrm{CF}-\mathrm{MAF}$$

In it Δc_p_ stands for the absolute heat capacity of a semi-crystalline polymer, while Δcp,pure represents the same for a completely amorphous polymer at each glass transition. This ratio represents the mobile amorphous portion participating in the glass transition [14].
(2)$$\mathrm{MAF}=\frac{\Delta c_{p}}{\Delta c_{p,\mathrm{pure}}}$$

The crystalline portion is calculated from the melt enthalpy Δh_m_ as well as the enthalpy Δh_cc_, which occurs during a possible post-crystallization. Δh_0_ is the theoretical enthalpy that must be applied to melt a 100 % crystal [13].
(3)$$\mathrm{CF}=\frac{\Delta h_{m}-\Delta h_{\mathrm{cc}}}{\Delta h_{0}}$$

The rigid amorphous phase contributes to the properties of a polymer. For example, the stiffness can be increased slightly because the rigid macromolecules behave similar to the crystallized ones. The opposite is true for the diffusion properties. While the crystalline regions prevent diffusion, the rigid amorphous regions do not represent a good barrier due to their high free volume and in this case are more likely to be assigned to the mobile amorphous phase [13].

Fillers up to a certain amount can act as nucleating agents for crystallization, which can change the internal structure of polymers [13]. If the addition is too high, this can result in an impediment to crystallization. The reason for this is that the particles or agglomerates can no longer be incorporated into the crystal areas due to their size [12,13,14,15,16].

The described three-phase polymer system can be extended to a four-phase polymer system by compounding with the additive. In this case, the additive represents the fourth phase. The particles of the additive can also be surrounded by a rigid amorphous phase, as shown in Figure 2. This depends on any interaction between the polymer and the additive.

The gyration radius of a PLA is approximately 21 nm [17], which theoretically calculates the maximum distance between the particles to be approximately 84 nm, so at that point interaction between the particles would be expected. Equally distributed theoretical distances in Table 1 suggest that direct interaction of the particles is not always possible because they are not sufficiently close together at the filler levels used. Especially at lower filler levels, the distance is actually too big. For this reason, it can be assumed that the macromolecules play a crucial role in forming a flow inhibition, since the particles are simply too far apart to allow an interaction.

It is certainly the case that particles can sediment in the molten state. However, interaction between the sedimented particles or the agglomerates would then still be possible. This would reduce the interaction area, but a qualitative statement would still be possible at high filler levels.

The main objective is to get a better insight into highly filled plastics that show a flow restriction. Therefore, it is essential to know where the inner structure that reduces flowability comes from. This investigation focuses on the influence of the macromolecules as a crucial factor for polymer–polymer interaction that promotes a yield stress.

In particular, the mobility of the individual macromolecules should be investigated, since they can only be part of an inner structure if they are severely limited in their mobility. For this purpose, rheometry is used to illustrate the changed flow behavior, but without discussing the curves in detail. The flow behavior itself is influenced by many factors, such as the difference in density of the particles, which can cause particles to settle in the melt. Agglomerates can also form and thus influence the flow behavior, as can also be caused by the affinity between the polymer and the particles. However, this is the content of countless studies in the literature based on the use of rheometry. This interpretation will then be carried out with the aid of thermal analysis. The consideration that the RAF softens only when the crystalline areas melt is part of the theory and was also already so included in the considerations of the tie molecules. Although Wunderlich first reported this phenomenon in 1985 [8], it has come back into focus in recent years, partly due to improved measurement technology. Proof of the method is the subject of current literature. Nevertheless, measurement uncertainties or errors always occur and cannot be excluded in any measurement. However, it should also be noted that the present manuscript is not a method for the quantitative determination of the bonding of the polymer chains to the additive, but a qualitative statement.

## 2. Materials and Methods

For the compounding of highly filled plastics, the biopolymer Ingeo 4032D (PLA) from NatureWorks, Minnetonka, MN, USA, a type for blow molding and Polypropylene HP-400H, a type for thermoforming by Lyondell Basell, Frankfurt, Germany, served as matrix materials, which differ in their polarity. They were mixed with the mineral fillers in different proportions varying form 20 vol.% up to 50 vol.%. The fused silica Sikron SF800 (SK) and SF810-10/1 (SN) were given by TheMineralEngineers, Frechen, Germany.

In both cases, no surface modification was applied. Both particles are almost round and thus show a low aspect ratio. SN has the significantly smaller average particle diameter d_50_ with 0.5 µm compared to SK with d_50_ with 2 µm. The smaller average particle diameter has a larger surface area and leads to a higher tendency of the individual particles to agglomerate.

The melt mixing was carried out by means of a twin-screw extruder EBVP 25 of the manufacturer OMC, Cassina de ‘Pecchi MI, Italy. The rheological examination of the compounds was carried out with a rotary rheometer SR-200 of the manufacturer Rheometrics, Munich, Germany, in a plate–plate arrangement. The plate diameter was 25 mm at a measuring gap height of h = 1 mm and a temperature of ϑ = 210 °C for PP and ϑ = 180 °C for PLA. The measurement was shear stress controlled.

The capillary rheometer used was an R47 from Göttfert, Buchen, Germany. The melting of the plastic granules is accomplished by a connected single-screw extruder. The plastic melt is forced by the piston of the capillary rheometer through the nozzle shown schematically in Figure 3. To produce a concrete yield point, a horizontal double flat slit nozzle was used. The entry point of the melt into the nozzle is somewhat decentralized, resulting in two differently long flow paths.

Due to the identical cross-section geometry, the inlet pressure loss of the nozzles is cancelled out (gap height H = 0.855 mm to gap width of B = 12 mm).
(4)$$\Delta p_{\mathrm{long}}=\Delta p_{\mathrm{short}}$$

The lengths of the individual flat slots are LLong = 55 mm or LShort = 44 mm. The total length can, therefore, be written as
(5)$$L_{\mathrm{long}}{=L}_{\mathrm{short}}+\Delta L,$$
whereby the pressure gradients result in
(6)$$p_{\mathrm{long}}^{\prime }<p_{\mathrm{short}}^{\prime }$$

For the same gap heights selected here, this means for the shear stresses
(7)$$\tau_{\mathrm{long}}^{\prime }<\tau_{\mathrm{short}}^{\prime }$$

Due to the gradual reduction of the piston feed, the shear stress τ becomes smaller and smaller until τ = τ_0_ (i.e., the yield stress) is reached.

The plastic melt then stagnates in the long nozzle, while it still flows in the short nozzle [19]. It is assumed that at this point an internal network has formed which resists the pressure of the piston and thus prevents the plastic melt from flowing.

The feed rate of the plunger is not reduced further, but the pressure is maintained by it while the melt begins to cool down. Once the nozzle has cooled sufficiently, it can be opened and the solidified melt removed from the flow paths of different lengths. Since cooling has taken place under pressure, it can be assumed that the plastic melt has solidified in the yield point state.

Differential scanning calorimetry (DSC) was used to perform the thermal analysis of the compounds. The DSC instrument was a Netzsch DSC 204, Selb, Germany. The heating rate of 10 K/min was used and the samples were heated from room temperature to ϑ = 180 °C. Cooling was at 10 K/min, after which a second heating cycle was run from ϑ = −40 °C starting temperature to ϑ = 180 °C. The initial weight was 15 mg for all samples due to the high filler content. The amorphous fraction was determined analogously according to [20] by measuring Δc_p_ at the glass transition temperature.

The measurements of the filler content are performed by means of a thermal gravimetric analysis (TGA) 850 of Mettler Toledo, Columbus, Ohio, USA. The initial temperature is set at 200 °C, continuously increased to 500 °C at five Kelvin per minute and maintained there. The upper temperature limit is set at 500 °C, as preliminary tests have shown that the unfilled plastics are completely degraded, but the additives are not damaged. The result is the additive content in the sample which is important for the later calculation of the different proportions within the polymer.

## 3. Discussion

Figure 4 shows the complex shear viscosities of both plastics for the larger particles with different degrees of filling as a function of the angular frequency. In particular, at low angular frequencies, the complex viscosity increases sharply with the filler content. For PLA filled with 50 vol.% for PP respectively with 40 vol.% of SK, an approximately linear increase in the complex viscosity with decreasing angular frequency is observable in double-logarithmic representation. This indicates a strong flow inhibition even at high angular frequencies. At this high degree of filling, a zero viscosity is not detectable in the measured frequency range. However, it cannot be assumed that there is none, but it is very likely. Hao et al. have attributed the disappearance of zero viscosity to the formation of a particle network for compounds made from PLA and SiO_2_ [17].

In the present investigations, polarity has no significant influence on the flow behavior of the highly filled plastic melts which is the reason why they are not shown in this paper. The two polymers can be distinguished by the fact that PLA has polar end groups, whereas PP is completely non-polar. However, the PLA macromolecule is also partly considered non-polar due to the much larger methyl group, so Wen et al. were able to detect hydrogen bonds between the C=O double bond of PLA and the Si–OH group of silica nanoparticles, but this only generates a small physical bond [21]. Both SK and SN consist of more than 97 % SiO_2_. SiO_2_ is polar, but is present in crystalline form [22].

The behavior for the storage modulus in Figure 5 is even more obvious. It can be seen that the course of the storage modulus of the plastic melts with increasing filler content is independent of the angular frequency, which corresponds to the behavior of a solid. These courses can be determined equally for the two plastics. If the change in the storage modulus for low shear rates approaches zero, this indicates the formation of an internal network in the plastic [23,24]. It can, therefore, be stated that there is a formation of an inner structure in the plastic.

This formation of the inner structure is shown in exemplary form in a shear stress sweep of filled PP. In the range of low shear stresses, the course is linear-elastic, as can be seen in the Figure 6. Especially in this range there are strong fluctuations, because the deformations are very small and, therefore, the sensitivity of the measuring instrument reaches its limits. This is the reason why at low shear stresses there are sometimes no measured values available. For all three curves, however, a transition can be seen from which the deformation increases significantly. From this point on, the plastic melt shows a linear-viscoelastic behavior. This transition point is the yield point and is shifted towards higher shear stresses with increasing additive content. This can be traced back to the fact that the plastic melts with higher filling levels show increasingly solid-state behavior and thus behave linear-elastically. When the yield point is exceeded, the force acting from the outside is sufficient to break up the inner network [4].

This serves as further evidence of the formation of an internal structure.

The filler SN, with its smaller particle size compared to SK by a factor of four and thus a larger specific surface area, leads to significantly higher complex shear viscosities even at lower filler levels (Figure 7). The curves of the highly filled compounds proceed almost parallelly, whereby the influence of the degree of filling is clearly recognizable. The same applies to the storage modulus in Figure 8. It is noticeable that the influence on the complex viscosity with higher amount of the filler gets smaller with increasing filler content. Possibly, the particle structure is already so pronounced at that point that the tribological properties or contact interactions of the filler dominate [7]. Considering the small distances of the particles from a few to a few hundred nanometers, contact interactions in the form of friction, especially at higher filler levels, are very likely (Table 1). But a flow restriction occurs also for lower filler levels, when no interaction is expected.

The smaller SN particles lead to higher viscosity values than the plastics filled with SK particles. This is the case for the matrix made of PLA and of PP. Thus, it can be stated that the larger total surface area of the smaller particles has a stronger influence on the flow behavior of the melt.

Compared with [24], smaller particles lead to a stronger network, which explains the rapid increase in viscosity and thus the formation of the yield point. In general and as shown in Figure 1, the viscosity increases with smaller particles already by smaller addition amounts [17].

However, at these high fill levels it is likely that agglomerates will form. The literature also assumes that the particles can no longer be isolated from a certain degree of filling [25]. Although the smaller particles tend more because of their specific surface, this does not affect the formation of the inner network. This suggests that the larger surface area of the particles has a decisive influence on the formation of the internal structure.

A good possibility to infer the dispersion of additive particles is offered by [26]. A correlation between the rheological properties and the mechanical properties is derived. Here, too, the distance between the actual particles is theoretically determined, and then the influence of matrix and filler is distinguished from each other on the basis of the relaxation time spectra of the compounds and taken as a measure for the dispersion.

From this point on, flow behavior will not be discussed in detail. Rather, it should be noted as a partial result of comparison with the relevant literature that an internal network is formed.

The proportion of the rigid amorphous phase in Figure 9 and Figure 10 was calculated according to Schick [10], as shown in the Introduction. A value from the NATHAS database was not used as a basis, but a value for the plastics at hand was determined in preliminary tests without additives. A division of the polypropylene into a rigid and a mobile amorphous phase cannot be determined due to the rapid crystallization. The evaluation of the individual phases can be seen in Figure 9 and Figure 10.

This is just the proportion of the polymer, the amount of additive measured in TGA has already been excluded and subtracted. In the thermograms, which are not shown, it is very difficult to ever detect a glass transition point and to evaluate it reproducibly. Possible reasons for this are that particles not only act as nucleation sites but, if they are sufficiently large, can hinder the crystallization [14,15,27].

Both the cooling rate and the crystallization rate influence the size of the spherulites of the polymers. The results of the thermal analysis in terms of enthalpy of fusion are shown in Table 2. It can be generally said that a simpler structure of the macromolecule allows a higher crystallization rate [28]. Small spherulites, which in turn produce a large surface area, come from a high crystallization rate.

However, another statement can also be derived from this observation. Many small crystals produce a large surface area and thus a large rigid amorphous fraction, with many macromolecules being part of one or even more crystals. This in turn means that the mobile amorphous phase becomes very small, approaching zero.

This could be an explanation why no or only a small glass transition step can be measured for rapidly crystallizing polymers. The macromolecules are either completely or at least partially part of a crystal. This limits an evaluation of the flow behavior of polypropylene, but it can be inferred: Crystallization and rigid amorphous phase correlate. 

After cooling in the die, similar values are also measured during the second heating (see Table 2), so it can be assumed that there is no interaction between the polymer and the filler. If there was indeed an interaction, the macromolecules would be restricted in their mobility and would not be able to crystallize to the same extent.

The macromolecules must, therefore, first be freely mobile in the melt, which would not be possible if they were bound to the filler, since the formation of the rigid amorphous phase only takes place through crystallization. From this, in turn, it can be concluded that particle–particle interaction must have occurred, because a flow limitation or yield point could be detected in the rheological characterization.

At this point, it should be noted that crystallization in PP is inhibited for filler levels above 20 m.% on mineral fillers. This is due to the fact that the crystal lamellae form perpendicular to the surface of the additives. However, if the additive content is too high, their growth is restricted. As a result, the additives, irrespective of their size, no longer serve as nucleation nuclei, but as inhibitors for crystallization. Therefore, the term nucleation optimum can also be found in the literature [29]. Nevertheless, no restriction of crystallization can be detected in the present investigations, which will certainly have to be addressed in further work.

For the PLA in Figure 10, it can be seen that there is a strong formation of the crystalline phase. The polymer has not been damaged by extrusion in the capillary rheometer, leaving it completely amorphous after the second heating. However, all PLA compounds show a crystalline content, so that the distinction in a rigid amorphous phase, which is provoked by the filler, cannot be clearly made. Thus, there is a rigid amorphous fraction which surrounds the crystals [30].

However, no interactions between the filler and the polymer (no RAF) were detectable even after the compounding process in previous investigations when there was no further pressure applied by an external force. The macromolecules were all fully movable and by that part of the MAF [31]. Thus, it can be stated for the polylactide that the influence of the polymer on the flow behavior of highly filled plastics is negligible.

An indication that the result found is valid is provided by a comparison with copper as an additive, in which no bond between polymer and additive is to be expected.

A further proof of the great influence of the particles can be shown for the first time in pictorial form (Figure 11) by means of initial optical investigations on PP filled with copper platelets, since this has a good contrast between the polymer and the particles in comparison with silicon dioxide. That is, the particles are displaced relative to one another within the plastic melt by the application of an external force or pressure. In this case, this force is applied by the piston in the capillary rheometer up to the point where no more plastic melt emerges from the longer flow channel. However, the melt could still flow simultaneously from the shorter flow channel. It is then assumed that, due to the higher pressure drop in the longer melt channel, the resistance caused by the internal structure is sufficiently high so that the melt stagnates. These pictures are compared with pictures taken on strands after the compounding process. Here, the plastic melt is able to solidify at ambient pressure without external force and it can be assumed that no external force has acted on the sample. Figure 11 shows a direct comparison of the two states. It is clear that the particles are closer together.

## 4. Conclusions

For both mineral additives, flow inhibition could be clearly demonstrated by plate–plate rheometry, irrespective of the filler content. Since the macromolecules were freely mobile in the melt, as measured by thermal analysis, and did not detectably form a rigid amorphous phase even under force during cooling, interaction between particle and polymer is ruled out.

The internal structure, which ultimately results in the yield point, is thus formed without the influence of the macromolecules. Rather, the additives play the decisive role here, even if they are theoretically too far apart. This indicates that, in the state of the yield point, the spacing of the particles is changed, which could be reproduced by cooling under constraint. Indeed, initial images in the yield limit state show a displacement of the particles that reduces the spacing such that direct interaction of the particles becomes possible. The formation of a polymeric network can be ruled out due to the low rigid amorphous phase during initial heating. It can be assumed that this is caused by the high degree of crystallization. Further investigations of the relationships between the flow behavior and the thermal properties in the solid and their informative value are to follow.

## Figures and Tables

**Figure 2 polymers-13-03349-f002:**
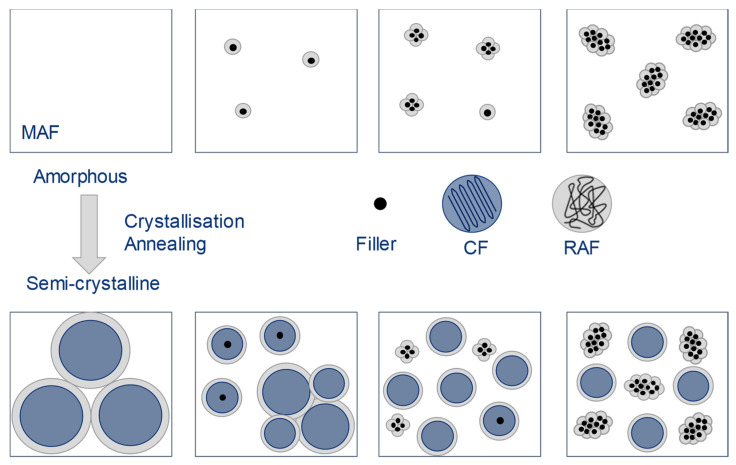
Rigid amorphous phase around crystalline domains and filler particles [12]. Depending on the amount added, the particles can come together to form agglomerates. An RAF can develop both as part of crystals and around additives.

**Figure 3 polymers-13-03349-f003:**
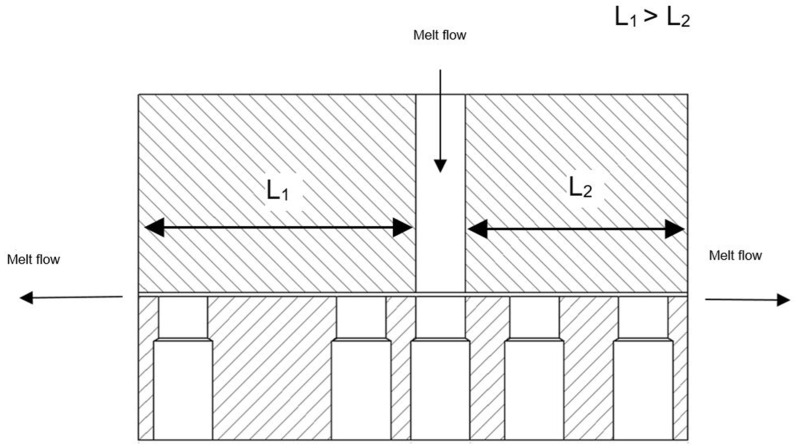
Die for the capillary rheometer with different melt path lengths [19]. By reducing the advance of the piston, it can be achieved that the melt stagnates on one side of the die while it is still flowing on the other side.

**Figure 4 polymers-13-03349-f004:**
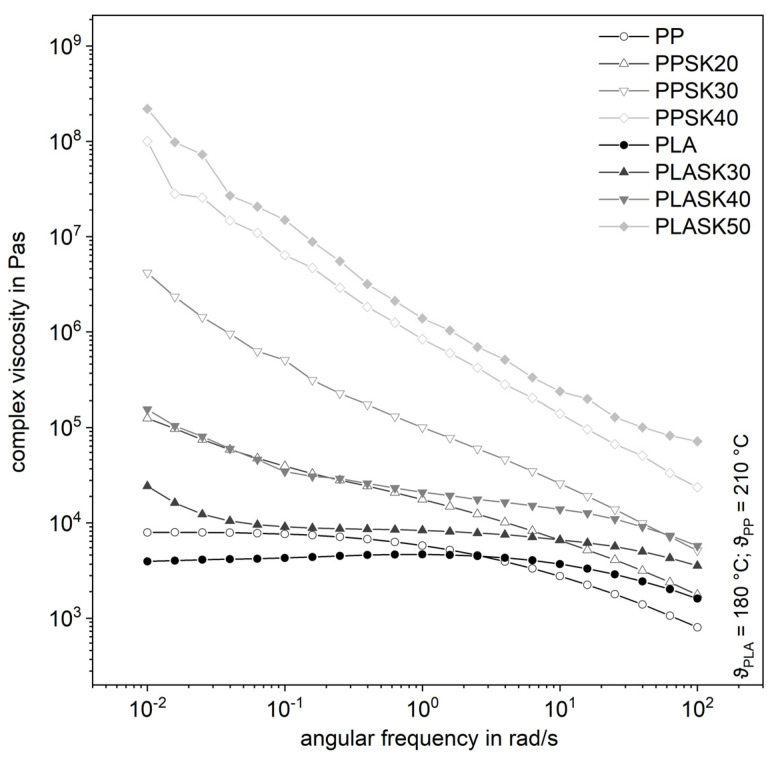
Influence of the plastic matrix on the complex shear viscosity of the compounds with the larger fused silica (SK).

**Figure 5 polymers-13-03349-f005:**
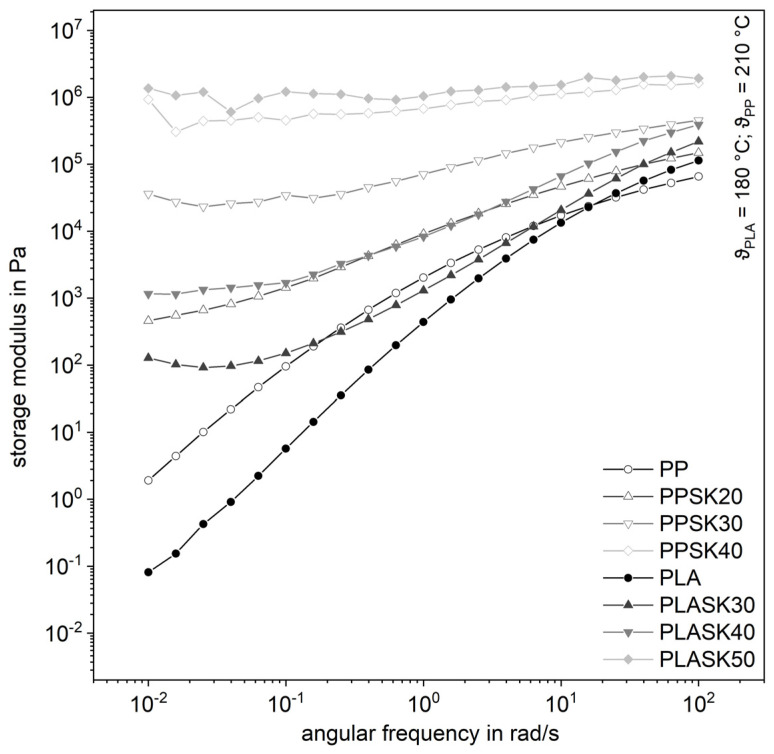
Influence of the plastic matrix on the storage modulus of the compounds for with the larger fused silica (SK).

**Figure 6 polymers-13-03349-f006:**
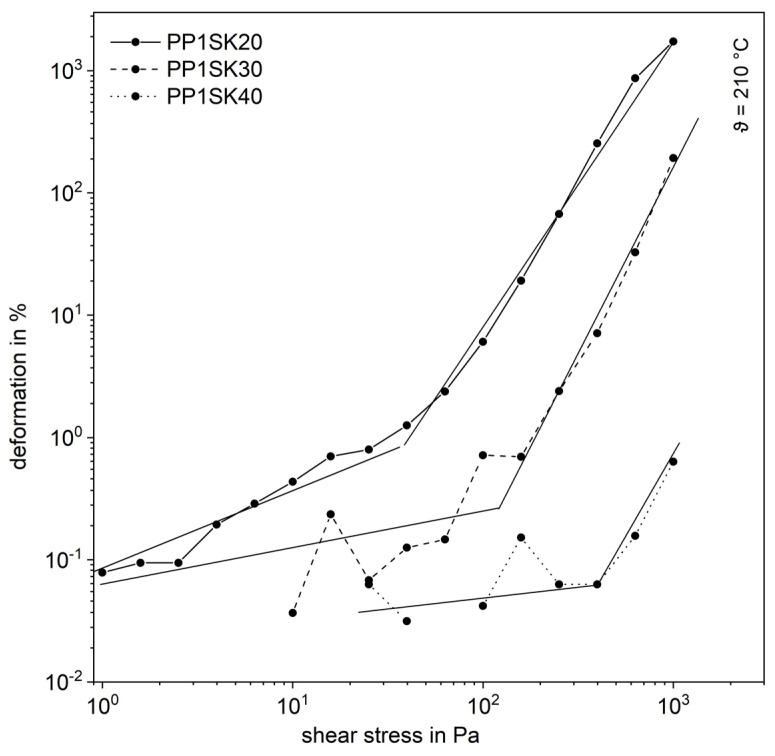
Shear stress sweep for the compounds with the larger fused silica (SK).

**Figure 7 polymers-13-03349-f007:**
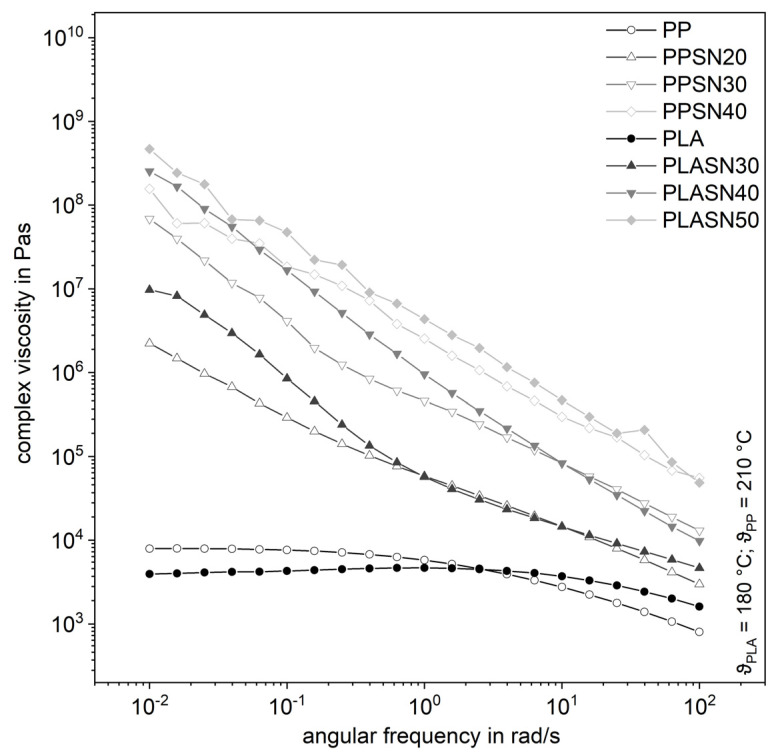
Influence of the plastic matrix on the complex shear viscosity of the compounds with the smaller fused silica (SN).

**Figure 8 polymers-13-03349-f008:**
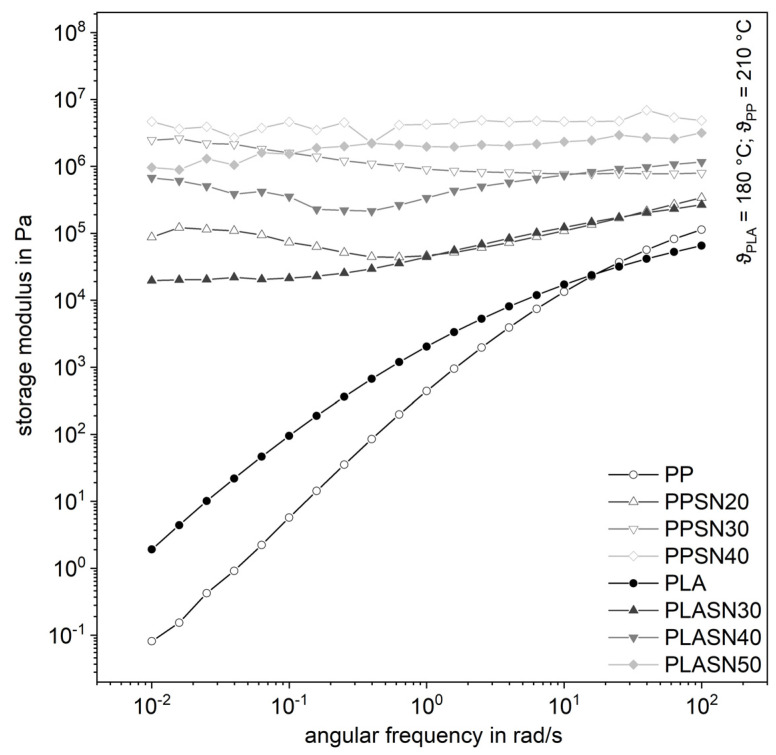
Influence of the plastic matrix on the storage modulus of the compounds with the smaller fused silica (SN).

**Figure 9 polymers-13-03349-f009:**
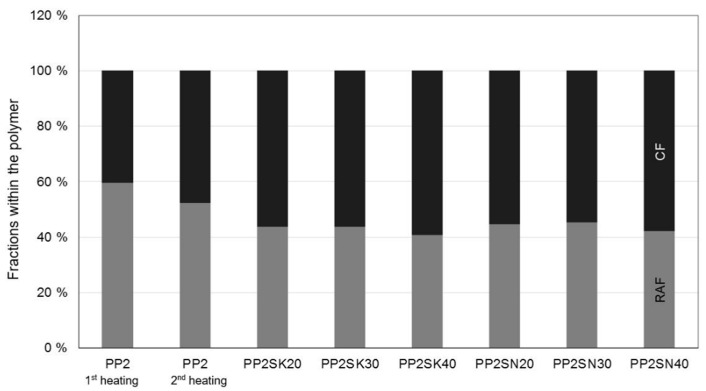
Different properties within the PP. SK is used for the larger particles, whereas SN describes the smaller particles (in %).

**Figure 10 polymers-13-03349-f010:**
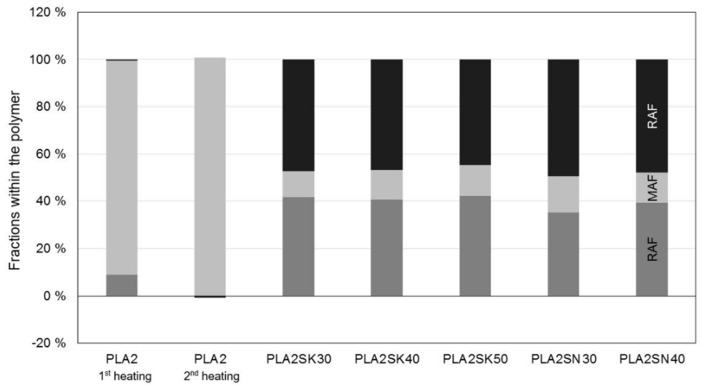
Different properties within the PLA. SK is used for the larger particles, whereas SN describes the smaller particles (in %).

**Figure 11 polymers-13-03349-f011:**
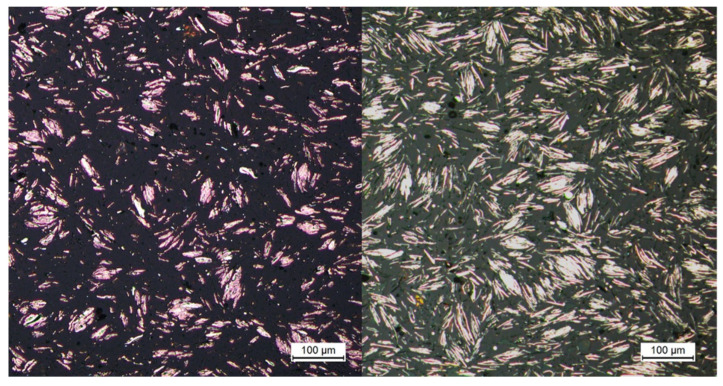
Cooling without external force (**left**), cooling with an external force produces by the piston of the capillary rheometer (**right**) for the same plastic.

**Table 1 polymers-13-03349-t001:** Theoretical distances between the individual particles in PLA calculated according to [18].

Filler Content in vol.%	Particle Size in µm	Particle Distance in nm
30	0.5	102
40	0.5	47
50	0.5	8
30	2	408
40	2	188
50	2	31

**Table 2 polymers-13-03349-t002:** Melt enthalpies for PP2 in the first and second heating processes.

Plastic	1. Heating		2. Heating	
	Δh_m_ in J/g	Δh_cc_ in J/g	Δh_m_ in J/g	Δh_cc_ in J/g
PP	83.640	0	98.870	0
PPSK30	67.920		64.400	
PPSK40	54.130		50.730	
PPSK50	40.580		37.560	
PPSN30	71.300		67.300	
PPSN40	55.970		51.370	
PPSN50	49.420		44.630	

## Data Availability

The data presented in this study are available on request from the corresponding author.

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
