# Peer review of "Rigid Amorphous Fraction as an Indicator for Polymer-Polymer Interactions in Highly Filled Plastics"

_polymers, 2021, doi:10.3390/polym13193349_

Round 1
Reviewer 1 Report
In the manuscript “Rigid Amorphous Fraction as Indicator for Polymer-polymer-interactions in Highly Filled Plastics”, the authors present a study on influence of polymer-polymer-interaction that promotes a yield stress, rheometry is used to illustrate the changed flow behavior. The study uses two polymers (PLA and PP) widely used today and of great interest in the manufacture of different articles. The study is very interesting. Consider the following observations:
- Although figure 1 is illustrative, place scale, and units.
- It is not indicated in the text, that the rheometry of PLA were made at 180 C and the PP at 230 C?
- The capillary rheology is widely described, and no results are presented.
- This investigation “focuses on influence of the macromolecules as a crucial factor for polymer-polymer-interaction that promotes a yield stress”, However, no results are presented nor is it discussed about yield stress.
Author Response
Dear Reviewer,
thank you very much for the review and the fact that you find the research interesting encourages me in my intention.
Regarding the comments:
1) The diagram is now described in more detail in the caption. However, since the viscosity is a relative value, no unit can be given here. The amount of additive added is in volume percent.
2) The information was added in the experimental procedure.
3 and 4)
This is correct. The capillary rheometer is used virtually only as a means to an end. The actual representation of the yield point has not been done. Rather, attention was paid here to the stagnation of the melt on one side of the nozzle. It can therefore not be ruled out with complete certainty that the melt was still slightly in motion. In future measurements, a correlation should be generated here which, in the very best case, can also be referred to the measurements of the plate-plate rheometry (verification of the Cox-Merz relation for filled plastics).
Reviewer 2 Report
In this manuscript authors investigated the rheological behavior of Polylactic acid (PLA)/SiO2and Polypropylene (PP)/SiO2 particulate composites (commercial PLA and PP) claiming the prediction of polymer-filler interactions on the base of the so-called polymer three phases theory . This theory was manly developed by B. Wunderlich and states that in semicrystalline polymers there is a rigid amorphous fraction ( RAF) not contributing to glass transition of the mobile amorphous phase. According to this theory, the rigid amorphous fraction has a high glass transition temperature (hardly detected experimentally) that can even overcome the melting point of the crystalline fraction. The present manuscript extends the three phase theory to composites, stating that the presence of interacting fillers increases the RAF because of the additional contribution of the interphase between the polymer and the filler. Since was not detected any increase of the RAF, authors argued that no interaction between matrix and filler exists and that the increase in viscosity of composites was due to interactions amongst particles of the filler.
Actually, this reasoning is not correct since a simple hypothesis (the increases of RAF in composites as a consequence of polymer-filler interactions) is considered by the authors as an axiom and used for deductive purposes. The scientific method instead has to be based on inductive reasoning and, therefore, all possible causes of the changes of viscosity have to be explored without prejudice .
Moreover, only foreign atoms or molecules can be incorporated in crystals, whereas particles are too large and can only act as substrates for crystallization because of their different thermal properties or, rarely, of epitaxy.
For this reasons, I do not recommend the publication of the manuscript in Polymers. If the authors want their manuscript to be published they have to limit the discussion to the rheology of the composites, to consider appropriate hypothesis on the base of the experimental data and carry out the right investigation to confirm or exclude them, avoiding inappropriate mentions to the three phase theory not only in the title but in the entire manuscript.
Author Response
Dear reviewer,
Thank you for your comments on the manuscript, which I am happy to address here.
Indeed, the idea and from it also the hypothesis have emerged from the observation of a phenomenon. In no way do the authors wish to derive an axiom or even a law here. Rather, the phenomenon of flow inhibition is to be examined in more detail. Especially filled plastics show an altered flow behavior up to a flow inhibition or limit, although the amount of the additive added is rather small. And it is so small that, theoretically, the particles cannot touch each other if they are uniformly distributed in the polymer. Therefore, the question arises to what extent an internal structure nevertheless forms and what role the polymer chains play in this. We exclude a pure network of only polymer chains, so one consideration was whether the polymer chains that bind to a particle form this internal structure. However, if there is no binding at all between the macromolecule and the additive, this cannot be the reason for the flow obstruction either. At the same time, the picture at the end of the manuscript in the state of flow inhibition clearly showed that the particles are displaced against each other by the pressure of the processing and thus can suddenly interact directly with each other. We expected this behavior, but such a pictorial representation has not been found in the literature so far.
The rheological investigations by means of plate-plate rheometer serve here for the comparison with the literature that a corresponding inhibited behavior occurred in the melt.
The remark on crystallization is correct, so that crystals are no longer referred to here, but crystalline areas.
Reviewer 3 Report
The authors present a manuscript relate to the rigid amorphous fraction study in high filled plastic. The paper is well written, readable and constist in all the necessary part. My only comment is to complete the figure in a better way: each one has to be self standing, a reader who does not want to o deeply in the reading has to undestand all the acronym etc only by reading the caption; figure 1 is well done, but the acronyms, for instance, that you have used in fig.2 have to be explained.
Author Response
Dear reviewer,
thank you very much for the review and the fact that you find the examine interesting encourages me in my intention. The captions are now more detailed, so the acronyms are explained here as well.
Round 2
Reviewer 2 Report
The so-called three phases model is based on the comparison between the difference ∆Cp = Cp,L-Cp,S (where Cp,L is the heat capacity of the liquid polymer and Cp,S the heat capacity of the solid polymer) determined by DSC at the glass transition temperature and the same difference ∆Cp calculated by Nathas database. The disagree between the " experimental and calculated" ∆Cp has been ascribed to the existence of a rigid amorphous phase (whose existence, to the best of my knowledge, has never been definitively proved and even leads to the paradox of a glass transition of RAF higher than the melting temperature of the crystalline phase ) without taking into account other reasonable explanations, including sources of errors and inaccuracy of both experimental and theoretical data. Referring to present manuscript, please also note that even if a good dispersion of ceramic particles in a solid polymer matrix is achieved, in a melted matrix particles can agglomerate because of their higher density and/or their scarce affinity to the polymer, modifying therefore the rheological behavior. Since the authors have ignored my comments and based also the revised version on the postulates of the three phases model, I have to reject once again the manuscript.
Author Response
The representation for the determination of the RAF is correctly described. The consideration that the RAF softens only when the crystalline areas melt is part of the theory and was also already so included in the considerations of the tie molecules. Although Wunderlich first reported this phenomenon in 1985, it has come back into focus in recent years, partly due to improved measurement technology. Proof of the method is the subject of current literature. Nevertheless, measurement uncertainties or errors always occur and cannot be excluded in any measurement. However, it should also be noted that the present manuscript is not a method for the quantitative determination of a bonding of the polymer chains to the additive, but a qualitative statement. The extension of the three-phase system to a four-phase system with additives is described in the literature. In preliminary tests with other additives that allow binding, the suitability of the method was demonstrated. Also, the theoretical quantity ∆cp from the NATHAS database was also not used, but referred to the pure polymer without additives, which was determined beforehand. Only then were further investigations triggered by means of the flat sheet die, which in turn is supposed to provoke a flow-inhibiting behavior. This condition was then again investigated in detail.
It is certainly the case that particles can sediment in the molten state. However, interaction between the sedimented particles or the agglomerates would then still be possible. This would reduce the interaction area, but a qualitative statement would still be possible at high filler levels.
But this was also a thought that led to the preparation of the manuscript. Not to approach the observation of the yield point phenomenon with rheological methods, because there are already a large number of studies on this in the literature, which deal precisely with these issues of agglomeration, affinity, geometry, etc., but to try to combine it with the method for determining the RAF, which has not yet been found in the literature in this way.